# Triple-Negative Breast Cancer Subclassified by Immunohistochemistry: Correlation with Clinical and Pathological Outcomes in Patients Receiving Neoadjuvant Chemotherapy

**DOI:** 10.3390/ijms25115825

**Published:** 2024-05-27

**Authors:** Bruno de Paula, Susanne Crocamo, Carlos Augusto Moreira de Sousa, Priscila Valverde, Fabiana Rezende, Eliana Abdelhay

**Affiliations:** 1Núcleo de Pesquisa Clínica, Hospital do Cancer III, Instituto Nacional de Câncer –, Rio de Janeiro 20560-121, Brazil; crocamo@gmail.com; 2School of Biosciences, Faculty of Health and Medical Sciences, University of Surrey, Guilford GU2 7XH, UK; 3Faculdade de Ciências Medicas, UERJ-Universidade do Estado do Rio de Janeiro, Rio de Janeiro 20551-030, Brazil; 4Divisão de Patologia, COAS, Instituto Nacional de Câncer–INCA, Rio de Janeiro 20220-400, Brazil; 5Divisão de Laboratórios Especializados, COAS, Instituto Nacional de Câncer–INCA, Rio de Janeiro 202300-130, Brazil

**Keywords:** triple-negative breast cancer, immunohistochemistry, neoadjuvant chemotherapy

## Abstract

The intrinsic subtype of triple-negative breast cancer (TNBC) is based on genomic evaluation. In this study, we report the survival and pathological complete response (pCR) rates of TNBC patients subtyped by IHC and treated with neoadjuvant chemotherapy (NACT). A retrospective cohort of 187 TNBC patients who received NACT between 2008 and 2017 was used, and IHC subtyping was performed on biopsy specimens before chemotherapy. The subtyping revealed predominantly basal-like tumors (IHC-BL, 61%), followed by basal-like immune-suppressed tumors (IHC-BLIS, 31%), mesenchymal tumors (12.5%), luminal androgen receptor tumors (IHC-LAR, 12%), and basal-like immune-activated tumors (IHC-BLIA, 10.9%). The pCR rate varied among subtypes, with IHC-BLIA showing the highest (30.0%) and IHC-LAR showing the lowest (4.5%). IHC-BLIS led in recurrence sites. Overall and disease-free survival analyses did not show significant differences among subtypes, although IHC-BLIA demonstrated a trend toward better survival, and IHC-mesenchymal, worse. Patients who achieved pCR exhibited significantly better disease-free survival and overall survival than non-responders. This study underscores the potential of IHC-based subtyping in TNBC management, highlighting distinct response patterns to neoadjuvant chemotherapy and potential implications for treatment strategies. Further research is warranted to validate these findings and explore tailored therapeutic approaches for specific TNBC subtypes.

## 1. Introduction

According to Cancer Research UK, triple-negative breast cancer (TNBC) accounts for approximately 15% of malignancies affecting the breast. Characterized by the absence of estrogen, progesterone, and Her2 receptor positivity, it is usually associated with a poor prognosis, and chemotherapy remains the main basis of systemic treatment [1]. In the last decade, the relevant benefits of chemotherapy combined with immunotherapy and targeted therapy for patients with pathogenic germline BRCA mutations have positively impacted outcomes in both curative and palliative settings [2].

However, TNBC represents a heterogeneous group of several subtypes with distinct drivers, survival outcomes, and responses to systemic therapy [3]. Several classifications have been proposed and validated, largely based on genomic evaluation of the tumors [4]. In 2011, Lehmann et al. characterized seven subtypes by gene expression, including one unstable. Basal-like 1 (BL1) is associated with genomic repair pathway deficiencies; basal-like 2 (BL2) is enriched in growth factor signaling-related genes; immunomodulatory (IM) is enriched in immunological signaling; and mesenchymal (MES) and mesenchymal stem–like (MSL) are enriched in components and pathways related to cell motility and the luminal androgen receptor (LAR), and androgen and metabolism genes are enriched [5]. Later, in 2015, Burstein described four subtypes, namely, LAR, MES, basal-like immune-suppressed (BLIS), and immune-activated (BLIA) subtypes [6]. One year later, the FUSCC classification introduced the immune-modulatory (IM) subtype, along with the known MES, BLIS and LAR subtypes [7].

All the above classifications are based on molecular profiling, which could be valuable at the patient level for precision medicine approaches once some mutations predict the benefit of targeted therapy. However, for most patients, the treatment options will remain limited. Moreover, technical challenges, such as the need for fresh tissue, the length of time needed to perform the analysis, and the significantly high costs, impact its wide use in standard practice [8,9].

Immunohistochemistry (IHC) is an established laboratory method for evaluating the presence of antigens that is widely used in breast cancer and is pivotal for tailoring systemic treatments (e.g., the human epidermal growth factor receptor 2—HER2) [10]. Moreover, IHC is significantly less expensive and laborious than gene expression and has been evaluated in several studies as a surrogate marker of established intrinsic genomic-based subtyping assays, as summarized in Table 1 [11,12,13,14,15,16,17,18]. Although some studies have provided survival data amongst the subtypes, the effect of chemotherapy on the pathological complete response, and if it serves as surrogate biomarker of survival, has not been reported.

In this study, we report the survival and pCR rates of patients who underwent IHC subtyping using our panel and who received neoadjuvant chemotherapy.

## 2. Results

One hundred eighty-seven women were included in this analysis; the mean age was 52.43 (SD = 12.79) years, the mean tumor size was 47.64 (SD = 53.61) millimeters, and the patients’ characteristics are summarized in Table 2.

After balancing pros and cons from the antibodies used by studies presented in Table 1, we chose the following four for our analysis: IDO-1—Indoleamine 2,3-dioxygenase (IDO-1), Forkhead Box C1 (FOX-C1), claudin-3 [19], and androgen receptor. Staining for all 4 markers was possible in 98.4% (184s) of the patients and partially in 3 patients due to tissue exhaustion. We successfully identified the characteristics of 4 subtypes in 96.3% (183) of patients. Overall, the frequency of the IHC-basal-like (BL) subtype was 61% (114). The percentages of basal subtypes identified by IHC-BLIS and IHC-BLIA were 31% (57) and 10.9% (20), respectively. The second most common subtype was IHC-MES 12.5% (23), followed by IHC-LAR 12.0% (22) and IHC-mixed 11.4% (21) (Table 2).

Among the 28 (15%) patients who achieved pCR, 75% exhibited an IHC-BL, 14.28% exhibited an IHC-MES feature, and 7.4% exhibited an IHC-LAR feature. The pCR rate in the IHC-BLIA group was the highest (30%), and that in the IHC-LAR group was the lowest (4.5%) (*p* = 0.24) (Table 3).

Patients who achieved pCR had a significantly better median disease-free (mDFS) survival NR (not reached) (CI—confidence interval: >50%) vs. no-pCR 30 m (months) (CI: 18-NR) (*p* = 0.00022). Overall, it was also significantly greater in patients who achieved pCR according to the NR (CI: >50%) than in those who did not achieve pCR according to the 58-m (CI: 40-NR) no-pCR (*p* = 0.00018). (Appendix A).

Most of the recurrences occurred in the first 20 months following surgery. mDFS did not significantly differ among subtypes (*p* = 0.52). (Figure 1A) IHC-BLIA demonstrated a greater mDFS for patients with NR (CI: 34-NR) than for patients with all other subtypes (CI: 23-NR). The mDFS was also NR for the IHC-LAR, IHC-mixed, and IHC-BL groups. IHC-BLIS patients had the second worst mDFS, 50 m (CI: 11-NR), and IHC-MES patients had a markedly short mDFS, 16 m (CI: 6-NR). Interestingly, the mDFS curves for patients who did not achieve pCR were less distinct among subtypes, suggesting that pCR has a crucial role regardless of intrinsic subtype. IHC-MES followed by IHC-BL showed worse survival: respectively, 17 m (CI: 6-NR) and 15 m (CI: 6-NR) (*p* = 0.41) (Figure 1B).

Overall survival analysis also revealed no significant differences among subtypes (*p* = 0.61) (Figure 1C). IHC-BLIA had a greater mOS, NR (95% CI: NR > 50%) than the other subtypes at 60 m (95% CI: 48-NR) (*p* = 0.081). IHC-BLIA exhibit the graphically highest mOS, with an NR (95% CI: 39-NR) mOS for patients who did not achieve pCR and IHC-MES was associated with a worse survival with 37 m (95% CI: 27-NR) (*p* = 0.73) (Figure 1D).

The pattern of recurrence was graphically distinct among subtypes, although not statistically significant. IHC-BLIS led recurrence in each category, and interestingly, no local or cerebral relapse occurred in patients with the IHC-LAR subtype (Figure 2).

## 3. Discussion

This study is groundbreaking in terms of demonstrating pCR rates among TBNC subtypes using IHC. We found numerically higher pCR rates in the IHC-BLIA (30%) subgroup and a trend toward longer DFS and OS, while in the IHC-MES subgroup, survival was worse despite the pCR rate (17.4%) being closer to that of the whole population analyzed (15.0%). Achieving pCR was associated with longer disease-free and overall survival.

The basal-like subtype was predominant in our sample, with 61.5% of the tumors harboring a basal component. In 2010, Oakman, C. et al. reported higher rates of the BL phenotype in TNBC patients (71–91%) [20]. Interestingly, Lehmann et al. in 2011 reported that 49% of BL cases were characterized by intrinsic subtyping, but IHC seems to indicate a higher percentage (88%), and very likely, only half of these cases would correlate with the intrinsic BL subtype [5]. We observed rates somewhat closer to those reported by Zhao et al. and Jing Liang et al. [14,16] Identifying basal-like TNBC is clinically relevant, as data showed a benefit from adjuvant chemotherapy for patients with non-basal-like tumors and residual disease following surgery, despite the use of neoadjuvant chemotherapy [21,22].

We identified a predominance of IHC-BLIS (31.0%) over IHC-BLIA (10.9%) in our population, which likely contributed to the overall modest pathological response rate of 15.0%. Although our IHC-BLIS rates are somewhat comparable to the values found by the authors performing IHC subtyping, the overall percentage of patients with the IHC-BLIA subtype is markedly lower (Table 1). On the other hand, we found a numerically higher pCR rate in IHC-BLIA (30.0%) than the pCR rate in IHC-BLIS (14.0%), although not statistically significant. Based on recent publications, the new standard of care for early and high-risk TNBC patients includes immunotherapy alongside a taxane-platinum-based dose-dense chemotherapy, regardless of the presence of immune receptors/markers [23]. However, data are still needed to understand the efficacy of these regimens in TNBC patient subpopulations.

Immuno-inflammation occurred solely at the IHC-IM subtype according to Jing Lian et al. (2022), but a significant proportion of the other subtypes are immune-excluded, which raises the question of whether those subjects would benefit from immunotherapy in clinic [16].

IHC-MES was the second most common subtype, and numerically, it was associated with worse mDFS and mOS, which is consistent with the findings in the literature [13,14,15]. Unfortunately, the benefit of conventional adjuvant systemic treatment for this subtype is often poor, if any, given its specific molecular characteristics, which has motivated researchers to consider regimens mimicking sarcomatous disease [24]. Moreover, given the preclinical and clinical activity of mTOR, PI3K, SRC/ABL, and angiogenesis inhibitors, we believe that tailored treatment for this subtype could be warranted [2].

The IHC-LAR subtype had the lowest pCR rate (4.5%) and was the third most common subtype in our sample. A long-standing effort has been made to offer anti-androgen targeting for this subtype, with no translation in registrational approvals. The main challenges might include the unreliable behavior of AR as a driver in TNBC, compared to that of ER in HR+ disease, as well as the cutoff of AR intensity to trigger intervention and the high-bar endpoints chosen in clinical trials to evaluate the AR-targeting effect [25]. As mentioned above, the current best adjuvant treatment for patients with residual disease is capecitabine; however, for specific subtypes, such as LAR, the benefit is unclear. Therefore, to answer this question and by leveraging our knowledge of prostate cancer, where androgen deprivation therapy plus androgen receptor pathway inhibitors can be combined with docetaxel, we believe that a cohort of patients with IHC-LAR in a potential umbrella trial for intrinsic subtypes of TNBC should be considered to receive adjuvant capecitabine plus ARPI [26].

The pattern of recurrence we observed in our IHC-based subtyping cohort aligned with that expected in the literature (Table 1). Notably, brain relapse, a known independent factor of poor prognosis, occurred in 8% of the patients with IHC-MES, 14% in IHC-BLIS, and approximately 10% of those with the other IHC subtypes, but interestingly, no patients with IHC-LAR experienced brain relapse. There is a longstanding discussion on how to optimize relapse monitoring in high-risk breast tumors, but no clear consensus has been reached. Our data could assist in identifying patients with a greater risk of recurrence, which could deserve more individualized monitoring given specific patterns of disease relapse and therefore might benefit from brain and visceral imaging, as well as promising emerging techniques, such as circulating tumor DNA, in addition to the current recommended approaches [27].

Our study demonstrated a low percentage of unclassifiable (2.2%) and IHC-mixed (11.4%) compared to the other studies mentioned in Table 1. In a study conducted by Yoo in 2022, the majority of patients with IHC-unclassifiable had presented with mesenchymal subtype on genomic classification [15]. Interestingly, Zhao et al., Leeha et al., and Hu et al. did not report mixed subtypes by using their IHC panels [14,17,18]. We believe IHC-unclassifiable and IHC-mixed should be expected given the IHC assay limitations, and support the heterogeneous features of a significant proportion of TNBCs [28]. Although this population is smaller, we are conducting further research to elucidate how they should be handled, and, more specifically, the role of additional testing to better define this population.

Although the results of this study are promising, they should be interpreted in light of these limitations. Our retrospective design imposes challenges in survival analysis including selection and recall bias. The eventual changes in TNBC treatment protocols over the study period could introduce a confounding variable to the outcomes evaluated, as well as other variables not analyzed, such as comorbidities and genomic background. In the case of implementation, our proposed IHC panel will eventually increase in cost compared to the current standard of practice recommendation and could eventually cause delays in standard of practice results delivery. Moreover, there are several technical hurdles linked to standardizing IHC protocols for diagnostic protocols, including their validation and approval by regulatory bodies, which would be required before the panel we proposed is considered outside of a research environment. There is no consensus about the use of an IHC subtyping panel, and IHC staining was performed utilizing archival tumor tissue from 2008 onward, which could have impacted the performance of the assay and requires validation using an established method for intrinsic genomic subtyping.

## 4. Materials and Methods

### 4.1. Sample Selection and Outcome Definitions

Patients with localized or locally advanced TNBC who were treated with neoadjuvant chemotherapy followed by curative surgery at the same institution between January 2008 and December 2017 and who had available biopsy and surgical specimens met the inclusion criteria for this retrospective cohort.

From medical files, clinical and pathological data were collected to construct a database using an electronic case report form (Appendix A.).

For this study, TNBC was defined as estrogen and progesterone receptor < 1% and HER2-negative +1 or +2 without fish amplification. Pathological complete response (pCR) was defined as the absence of invasive cancer on surgical specimens from the breast and axilla (ypT0/Tis pN0). Disease-free survival (DFS) was defined as the time between the day of the cycle of one day of chemotherapy regimen administration until cancer recurrence or cancer-related death. Overall survival (OS) was defined as the time between the day of the cycle one day after chemotherapy regimen administration until death related to any cause. In the case of loss of follow-up, the appointment date at the NCI, which contains information about disease and survival, was used to censor the patient for the analysis.

### 4.2. IHC Analysis and Interpretation

Tissue samples were converted into histological sections on previously salinized slides. Evaluation of the cellular atypia pattern was performed by hematoxylin-eosin (HE) staining. The evaluated tumor area was selected by a pathologist, and the in situ regions within the invasive tumors were also delineated to allow a better evaluation of the stains. Tissue sections were immunoassayed with a Polymer Detection System (RE7150-K, Leica Biosystems Newcastle Ltd., Balliol Business Park West, Benton Lane, Newcastle Upon Tyne NE12 8EW, UK) according to the protocol established by the manufacturer. Primary antibodies were incubated with the tissues for 18 h at 4 °C at different dilutions determined by titration, as shown in Table 4.

The antibody panel was adapted from studies reported in Table 1 and interpretation of the singleplex IHC staining results was performed by the author F.R., as shown in Table 5.

The scoring was reviewed by the senior authors E.A., and the final scores agreed between F.R. and E.A. The digitalization was performed using the Aperio Scanner and a representative illustration of the staining is shown in Figure 3.

For positive controls, tissues suggested by the antibody manufacturer’s datasheets were used. The reaction was visualized using diaminobenzidine (DAB), followed by hematoxylin counterstaining. Negative controls were prepared without the primary antibody. The positivity of the staining was analyzed in ten random fields and determined by manual counting using ImageJ version 1.54 [29] software according to the preestablished equation below.
$$\frac{\sum_{m=1}^{10}Sc\;}{\sum_{t=1}^{10}Tc}\times 100=p$$
*Sc* represents the number of stained cells in each field, where *s* = [1, 2, ∙∙∙, 10]; *Tc* represents the total number of cells in each field, where *t* = [1, 2, ∙∙∙, 10]; and *p* represents the mean percentage of positivity.

### 4.3. Statistical Analysis

Continuous variables are shown as the mean and standard deviation, and categorical variables are shown as the total value and percentage of the total value.

The time to overall survival was computed as the time of diagnosis until death or the last patient contact, and the disease-free survival time was computed as the time of surgery until recurrence or last patient contact.

The Kaplan–Meier method was sed to estimate survival curves, and the log rank test was performed to evaluate whether there were differences in covariate levels via univariate analysis. The significance level adopted to reject the null hypothesis was alpha equal to 0.05. A bar plot was used to demonstrate the frequency of subtypes at each site.

## 5. Conclusions

We successfully demonstrated that an affordable and widely reproducible panel of IHC assays can be used to subtype TNBC. Moreover, we were the first to demonstrate the pCR rate among IHC subtypes.

## Figures and Tables

**Figure 1 ijms-25-05825-f001:**
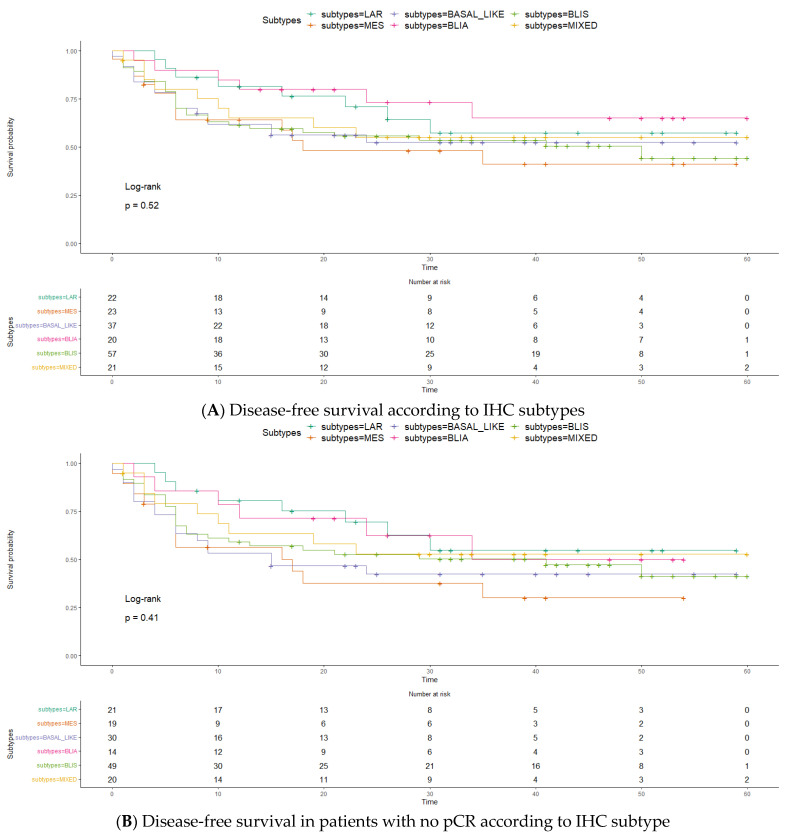
Legend: BLIS basal-like immunosuppressed; BLIA—basal-like immunoactivated; LAR—luminal androgen receptor; MES—mesenchymal. The time, expressed in months.

**Figure 2 ijms-25-05825-f002:**
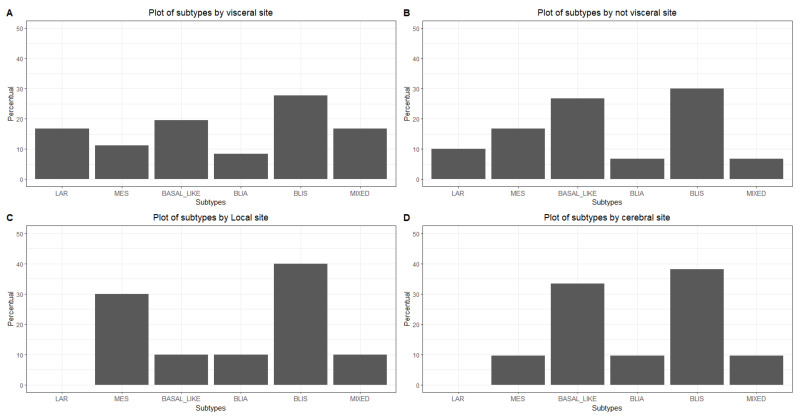
Pattern of recurrence among IHC subtypes. Legend: (**A**) Visceral recurrence; (**B**) Non-visceral/non-local/non-cerebral recurrence; (**C**) Local recurrence; (**D**) Cerebral recurrence. BLIS—basal-like immunosuppressed; BLIA—basal-like immunoactivated; LAR—luminal androgen receptor; MES—mesenchymal.

**Figure 3 ijms-25-05825-f003:**
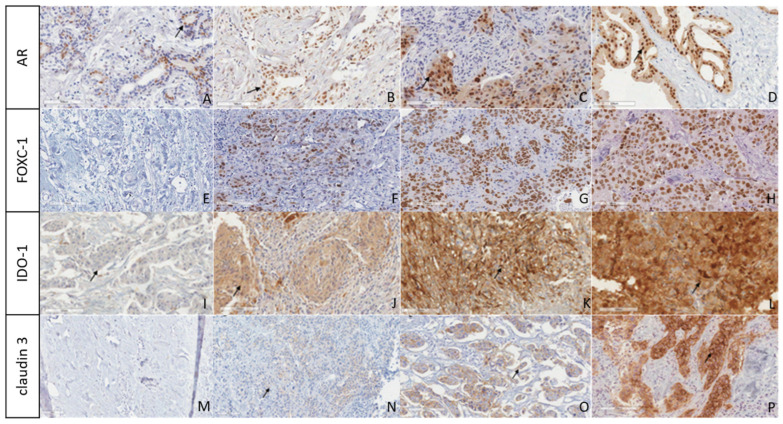
Immunostaining panel of tumor samples. Legend: Anti-AR immunostaining nuclear expression was observed in 10–30% (**A**), 31–50% (**B**), 51–75% (**C**), and 76–100% (**D**). Anti-FOXC1 immunostains were stratified according to nuclear staining intensity: negative (**E**), 4+ (**F**), 6+ (**G**), and 8+ (**H**). Anti-IDO1 immunostains were stratified according to the intensity of cytoplasmic staining in the tumor region as: ≤10% (**I**), 50% (**J**), 80% (**K**), and 100% (**L**). Anti-claudin immunostains were stratified according to the intensity of membrane staining in the tumor region as negative (**M**), weak (**N**), moderate (**O**), and intense (**P**). Magnification: 20×. Scale bar: 100 µm.

**Table 1 ijms-25-05825-t001:** Retrospective studies evaluating IHC as a surrogate for TNBC subtyping.

	Choi J et al., 2012 [11]	Kim S et al., 2018 [12]	Kumar S et al., 2020 [13]	Zhao S et al., 2020 [14]			Yoo T-K et al., 2021 [15]	Lian J et al., 2022 [16]	Leeha M et al., 2023 [17]	Hu H. et al. [18]
Total subjects	122	200	245	210	183	214	145	93	195	123
Methods used for subtyping	IHC	IHC	IHC	RNA and Gene expression and IHC staining			RNA and Gene expression and IHC staining	IHC	RNA and Gene expression and IHC staining	IHC
Adjuvant treatment	Chemo or radiation based on staging	NR	NR	~93% (taxane 75.7% and non-taxane 17.6%)	~90% (62.8% and 26.8%)	~91% (84.1% and 7.0%)	NR	NR	~ 90% (regimen not disclosed)	NR
IHC-Apocrine/LAR definition	AR and/or GGT1 > 10%	AR > 1%	AR ≥ 1%	AR ≥ 10%			AR Allred score 8 (5 + 3)	AR ≥ 10%	AR Allred score 8 (5 + 3)	AR ≥ 10%, regardless of the expression of other markers)
IHC-BLIS definition	NR	FOXC1 ≥ 4 and IDO-1 ≤ 10%	NR	AR–, CD8-, FOXC1 > 10%			NR	TIL low; AR < 10%; CD8 < 20%; FOXC1 ≥10%	NR	AR < 10%, CD8 TIL < 20%, FOX-C1 < 10%, and regardless of DCLK1 values
IHC-BLIA or IM definition	NR	IDO-1 > 10% and FOX-C1 < 4	NR	AR– and CD8 TIL activated ≥20%			LAR-negative and TIL score > 70%	AR <10%; TIL high; CD8 ≥ 20%	LAR-negative and TIL score > 70%	AR < 10%, CD8 TIL ≥ 20%, and regardless of FOX-C1 and DCLK1 values
IHC-Basal definition	CK5/6 > 10% and/or EGFR moderate or intense	CK5/6 and/or EGFR > 1%	BL1: EGFR < 4, CK5/6 ≥ 4 and/or CK4/14 ≥ 4	NR			LAR -, IM -, MES -, and with diffuse and strong p16 staining	NR	NR	CK5/6 and or EGFR+
			BL2: EGFR ≥ 4, irrespective of CK5/6 and/or CK4/14 result							
IHC-Claudin-low/Mesenchymal definition	Claudin 3, 4, 7 negative and/or e-cadherin negative	Claudin-3 negative and or e-Cadherin negative	E-cadherin, Claudin 3 and 7 ≥ 4, Vimentin ≥ 4	AR–CD8-FOXC1-DCLK1 ≥ 10%			LAR negative and TIL score < 20%	Metaplastic features; AR < 10%; CD8 < 20%; FOXC1 < 10%	LAR negative and TIL score < 20%	AR < 10%, CD8 TIL < 20%, FOX-C1 < 10% and DCLK1 ≥ 10%,
IHC-Mixed definition	2 characteristics of 2 different subtypes	2 or 3 different tumors	≥2 of other categories	NR			NR	NR	NR	NR
IHC-Unclassifiable definition	Not belonging to any subtype	None of the above features	Did not fit in any category	AR–CD8-FOXC1-DCLK1-			All other manifestations	Not reported	All other manifestations	AR < 10%, CD8 TIL < 20%, FOX-C1 < 10% and DCLK1 < 10%
IHC-LAR rate	12 (9.8%)	22 (11%)	41 (16.7%)	60 (28.6%)	42 (23%)	53 (24.8%)	26 (17.9%)	23 (24.7%)	37 (18.9%)	28 (28.6%)
IHC-BLIS rate	NR	11 (5.5%)	NR	80 (38.1%)	71 (38.8%)	90 (42.1%)	NR	39 (41.9%)	103 (52.8%)	20 (20.4%)
IHC-BLIA or IM rate	NR	27 (13.5%)	NR	40 (19.4%)	34 (18.6%)	39 (18.2%)	21 (14.5%)	24 (25.8%)	34 (17.4%)	39 (39.8%)
IHC-Basal rate	27 (22.1%)	85 (42.5%)	BL 36 (14.6%); BL1 32 (13.1%); BL2 4 (1.6%)	120 (57.1%)	105 (57.4%)	129 (60.3%)	BL1 27 (18.6%)	63 (67.7%)	137 (70.2%)	
IHC-Mesenchymal rate	28 (23%)	23 (11.5%)	70 (28.6%)	16 (7.6%)	18 (9.8%)	17 (7.9%)	44 (30.3%)	7 (7.5%)	1 (0.5%)	11 (11.2%)
IHC-Mixed rate	23 (18.9%)	60 (30%) LAR + MES 8 (4%); LAR + BL 27 (13.5%); MES + BL 19 (9.5%); LAR + MES + BL 6 (3%)	37 (15.1%)	0	0	0	NR	0	NR	NR
IHC-Unclassifiable rate	32 (26.2%)	10 (5%)	61 (24.9%)	14 (6.7%)	18 (9.8%)	15 (7%)	18 (12.4%)	0	20 (10.2%)	25 (20.3%)
Confirmation method	no	no	no	mRNA			mRNA	no	mRNA	no
follow-up median	59.5 months						41 m (0–64)	40 m (12–58)	40.95 m (IQR 23.48–89.22)	62 m (IQR 43–105)
Disease-free survival (DFS)	Basal-like and unclassifiable show less favorable prognosis, mesenchymal and mixed intermediate and AR showed a better prognosis Ck 5/6 and Claudin positivity worse DFS	BLIS—worse prognosis LAR, BLIA, BL and NOS—favorable This was also true for Burstein (4 subtypes). FOXC1–worse prognosis	NA	IM (HR = 0.07), LAR (HR = 0.18), BLIS (HR = 0.26) better RFS than MES	MES worse RFS	NA	Significantly worse DFS for M subtype according to surrogate subtypes and although not clinically significant IM tends to be better survival	No significant difference Low recurrence 11 cases (11.83%)	5 y 64.7% and no subtypes difference	IM-inflamed better DFS compared to others and BLIS the worse survival
Overall Survival (OS)	Cohesion disruption linked with worse survival	NA	Mesenchymal and unclassified shorted OS (68.2 and 69.2 m)	T + N + IHC was superior to T + N categories in time dependent AUC	NA	NA	NA	NA	5 y OS = 65.0% IM significantly better OS but other did not differentiate between each other	IM-inflamed better breast specific survival and BLIS the worse compared to others

Legend: IHC—immunohistochemistry; LAR—luminal androgen receptor; AR—androgen receptor; RNA—ribonucleic acid; GGT1—gamma-glutamyl transferase 1; FOXC1—Forkhead Box C1; IDO-1—indoleamine 2,3 dioxygenase-1; CD8—cluster of differential 8; TIL—tumor infiltrating lymphocytes; CK—cytokeratin; EGFR—endothelial growth factor receptor; IM—immunomodulatory; M—mesenchymal; DCLK1—doublecortin like kinase 1; MES—mesenchymal; T—tumor; N—lymph node; AUC—area under the curve; IQR—interquartile range; NA—not applicable; NR—not reported.

**Table 2 ijms-25-05825-t002:** Clinical and pathological summary.

**Variables**				**Mean**	**SD**
Age				52.42	12.79
Tumor size				47.64	53.61
Number of positive lymph nodes (Mean (SD))				2.14	4.01
ki67 (Mean (SD))				65.47	24.49
**Variables**				**Number of Subjects**	**Percentage**
Clinical tumor(T) stage	T1–T2			58	31.0%)
	T3			57	(30.5%)
	T4			72	(38.5%)
Clinical lymph node (N) stage	N0			79	(42.2%)
	N1			78	(41.7%)
	N2			28	(15.0%)
	N3			2	(1.1%)
Histologic subtype	Invasive ductal carcinoma			176	(94.2%)
	other			10	5.3%
	NA			1	0.5%
Tumor grade	1			2	1.1%
	2			59	31.6%
	3			118	63.1%
	NA			8	4.8%
Angiolymphatic invasion	Yes			98	52.4%
	no			39	20.9%
	Not-assessed			50	26.7%
Lymphatic infiltrate	Yes			32	17.1%
	No			100	53.5%
	Not-assessed			55	29.4%
IHC-subtype	Basal-like		Unspecified	36	19.6%
			Immunosuppressed	57	31.0%
			Imunoactivated	20	10.9%
	Luminal androgen receptor			22	12.0%
	Mesenchymal			23	12.5%
	Mixed			21	11.4%
	Unclassifiable			4	2.2%
	Not amendable of subtyping			3	1.6%
Neoadjuvant chemotherapy	ACx4 + TXTx4			144	77%
	ACx4 + wPacx12			19	10.2%
	TCx4			5	2.7%
	Other regimens			19	10.2%
Surgery modality	Modified radical mastectomy			120	64.2%
	Conventional mastectomy			17	9.1%
	Conservative surgery			50	26.7%
Pathologic response	Complete			28	15.0%
	Non-complete			159	85.0%
Pathologic complete response according to IHC-subtype	Basal-like	Unspecified		7	19.4%
		Immunosuppressed		8	14.0%
		Imunoactivated		6	30.0%
	Luminal androgen receptor			1	4.5%
	Mesenchymal			4	17.4%
	Mixed			1	4.8%
	Unclassifiable			1	25.0%

Legend: SD—standard deviation; N—number of patients; AC—anthracycline + cyclophosphamide; TXT—docetaxel; wPac—weekly paclitaxel; TC—docetaxel + platinum; x4—4 cycles; x12—12 cycles.

**Table 3 ijms-25-05825-t003:** Summary of the characteristics of patients who achieved a pathological complete response.

Subject Number	Age	Clinical Stage (TNM)	Histological Subtype	Grade	Ki67(%)	Neoadjuvant Regimen (x Number of Cycles)	Subtype
Patient 1	66	T4N1M0	IDC	2	90	cx3 + wPacx3	IHC-BLIS
Patient 2	63	T3N0M0	IDC	3	5	ACx4 + TXTx4	IHC-unclassifiable
Patient 3	54	T3N1M0	IDC	3	70	ACx4 + TXTx4	IHC-MES
Patient 4	67	TxN2M0	IDC	NR	70	ACx4 + TXTx4	IHC-LAR
Patient 5	47	T3N0M0	IDC	3	95	ACx4 + TXTx4	IHC-BLIS
Patient 6	60	T4N2M0	IDC	3	80	ACx4 + TXTx4	IHC-BL-unspecific
Patient 7	47	T4N1M0	IDC	3	80	ACx4 + TXTx4	IHC-BL-unspecific
Patient 8	52	T4N1M0	IDC	3	90	ACx4 + TXTx4	IHC-LAR/BLIS
Patient 9	56	T3N0M0	IDC	2	50	ACx4 + TXTx4	IHC-MES
Patient 10	44	T3N1M0	IDC	3	95	ACx4 + TXTx4	IHC-BL-unspecific
Patient 11	61	T4N2M0	IDC	2	70	ACx4 + TXTx4	IHC-BLIA
Patient 12	39	T3N0M0	Other	NR	95	ACx4 + TXTx4	IHC-BLIS
Patient 13	47	T2N0M0	IDC	2	95	ACx4 + TXTx4	IHC-BLIA
Patient 14	46	T4N3M0	IDC	2	60	ACx4 + TXTx4	IHC-BL-unspecific
Patient 15	44	T3N0M0	IDC	3	90	ACx4 + TXTx4	IHC-BLIS
Patient 16	68	T4N0M0	IDC	2	75	ACx4 + TXTx4	IHC-BLIA
Patient 17	64	T4N2M0	IDC	3	70	TCx4 + ACx6	IHC-BL-unspecific
Patient 18	41	T3N0M0	IDC	3	70	ACx4 + TXTx4	IHC-BLIA
Patient 19	63	T4N0M0	IDC	2	15	ACx4 + TXTx4	IHC-BLIA
Patient 20	34	T4N0M0	IDC	3	100	ACx4 + TXTx4	IHC-BLIS
Patient 21	37	T3N0M0	IDC	3	50	ACx4 + TXTx4	IHC-BLIS
Patient 22	71	T2N1M0	IDC	2	50	ACx4 + wPacx4	IHC-BLIA
Patient 23	43	T3N0M0	IDC	3	70	ACx4 + TXTx4	IHC-BLIS
Patient 24	47	T2N0M0	IDC	2	5	ACx4 + TXTx4	IHC-MES
Patient 25	62	T1N0M0	IDC	3	80	ACx4 + TXTx4	IHC-BL-unspecific
Patient 26	51	T2N1M0	IDC	3	80	ACx4 + TXTx4	IHC-MES
Patient 27	27	T2N0M0	IDC	3	95	ACx4 + TXTx4	IHC-BLIS
Patient 28	59	T4N0M0	IDC	3	60	ACx4 + wPacx4	IHC-BL-unspecific

Legend: T—tumor; N—lymph nodes; M—metastasis; IDC—invasive ductal carcinoma; NR—not reported; AC—anthracycline + cyclophosphamide; TXT—docetaxel; wPac—weekly paclitaxel; TC—docetaxel + platinum; x4—4 cycles; x3—3 cycles; IHC—Immunohistochemistry; BLIS—basal-like immunosuppressed; BLIA—basal-like immunoactivated; LAR—luminal androgen receptor; MES—Mesenchymal.

**Table 4 ijms-25-05825-t004:** Antibodies used in the analysis.

Antibody	Definition of Positivity Expression	Brand	Dilution	Control
IDO-1	>10% of tumor cells	Abcam *, Cambridge, UK	1:1000	Breast
Claudin-3	Allred score ≥ 4	Abcam *, Cambridge, UK	1:200	Breast
Androgen Receptor	≥20% of tumor cells	Cell Marque™, California, US	1:400	Gallbladder
FOX-C1	Moderate (M) or intense (I)	Abcam *, Cambridge, UK	1:300	Breast
Ki-67	≥1% of tumor cells	Ventana^®^-Roche, Rotkreuz, Switzerland	Ready to use	Breast

Legend: IDO-1—Indoleamine 2,3-dioxygenase, FOX-C1—Forkhead Box C1. * abcam is a registered trademark of Abcam Plc.

**Table 5 ijms-25-05825-t005:** Immunohistochemistry analysis and interpretation.

IHC-Subtype			Antibody Results			
			IDO-1	FOX-C1	AR	Claudin-3
Basal-like unspecific			IDO-1 ≤ 10% and FOX-C1 < 4 or IDO > 10% and FOX-C1 ≥ 4		<10%	W, M or I
Basal-like immune-suppressed			≤10%	≥4	<10%	
Basal-like immune-activated			>10%	<4	<10%	
Luminal androgen receptor			≤10%	<4	≥10%	
Mesenchymal			any	any	any	0
Mixed (criteria for ≥ 2 subtypes)	BLIS	LAR	≤10%	≥4	≥10%	W, M or I
	BLIA	LAR	>10%	<4	≥10%	
	BL	LAR	IDO-1 ≤ 10% and FOX-C1 < 4 or IDO > 10% and FOX-C1 ≥ 4		≥10%	
	MES	LAR	any	any	≥10%	0
Unclassifiable			≤10%	≤2	<5%	W, M or I

Legend: IHC—immunohistochemistry; BL—basal-like; BLIS—basal-like immunosuppressed; BLIA—basal-like immunoactivated; LAR—luminal androgen receptor; MES—mesenchymal; W—weak; M—moderate; I—intense; IDO-1—indoleamine 2,3-dioxygenase (IDO-1), FOX-C1—Forkhead Box C1; AR—androgen receptor.

## Data Availability

Authors are open to discuss data sharing on the terms of the National Cancer Institute.

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
