# Peer review of "Triple-Negative Breast Cancer Subclassified by Immunohistochemistry: Correlation with Clinical and Pathological Outcomes in Patients Receiving Neoadjuvant Chemotherapy"

_ijms, 2024, doi:10.3390/ijms25115825_

Round 1

Reviewer 1 Report

Comments and Suggestions for Authors

The study ijms-3001393 is a retrospective analysis that integrates the Lehmann classification system with a novel IHC panel. It examines the outcomes in terms of survival rates and pathological complete response rates in triple-negative breast cancer patients undergoing neoadjuvant chemotherapy, stratified by the new IHC panel. This initial investigation successfully categorizes response patterns and lays the groundwork for larger-scale studies.

Comments: 

1. The study acknowledges several limitations, including the retrospective nature of the design, challenges in survival analysis, lack of standardized IHC panels, and potential technical issues with older sample analyses.
However, the authors should rephrase these limitations to address the following: (a) Retrospective study designs are susceptible to biases such as selection and recall bias, complicating the establishment of causal relationships between variables; (b) Changes in TNBC treatment protocols over the study period (2008-2017) could introduce confounding variables that impact the study outcomes; (c) other confounding variables like comorbidities and genetic background or mutations are not addressed and can obscure the true effects of treatment interventions on outcomes. (d) Please also address the limitation in terms of delay and additional costs.

2. The authors have the opportunity to provide perspectives (author's views), suggest courses of action, or present considerations to readers regarding cases that defy classification. How should such cases be managed within the existing framework? The authors should elaborate on the perspectives and future directions given their results. The authors should specify if they are currently conducting further research on this subject.

For improvement:

3. Enhance the presentation of tables:
a. Table 1: Improve the readability. Populate empty cells with relevant information (e.g., NC, NA), for instance for the confirmation method for Choi 2012 and the definition of IHC-mixed for Yoo 2021.
b. Table2: Ensure consistency in the use of brackets and capitalization
c. Table5: Consider color improvement for readability.

4. Address issues in figures:
a. Include relevant p-values in Figure 1 as indicated in Line 143.
b. Indicate frequencies (in %) instead of absolute numbers in Figure 2.
c. Correct panels C and D to represent LAR in Figure 2 (refer to Line 200).
d. Specify missing details such as p-values and comparison groups (as highlighted in Line 145). Refer also to comment 5.d. regarding multi-testing.

5. In the text:
a. Specify gender in Line 79 or Table 4.
b. L147: Clarify whether staining was singleplex (or multiplex?).
c. L267: Provide information on the field of view (indicate: mm² and magnification power) for the selected images, and explain the randomization process (unsupervised?). Digitalization method of the sections is not indicated, please specify whether it was using validated Dx scanner.
d. Specify if the data followed normal distribution and homoscedasticity for conducting statistical analyses based on mean and standard deviation.
L276: Clarify if post-tests were conducted and if p-values were adjusted for multiple comparisons.

6. Address Missing Information:

a. Ensure supplementary materials are included (see Lines 111, 229, and 293).
b. Include representative staining illustrations in the main document and provide illustration of the controls in the supplementary data.
c. A pertinent reference on claudin-3 immunohistochemistry for TNBC (10.1186/s12885-018-4141-z) is missing.

Comments on the Quality of English Language

ImageJ is in the public domain and not a registered trademark. However, proper referencing to the software or the relevant plugin is necessary. Include a reference for the equation in Line 273.

1. Improve English grammar throughout the manuscript (required).
2. Correct spelling and typos like "discohesiveness", "alfa", "meDFS", "et all",...
3. Ensure consistency in the use of abbreviations, such as abbreviating Basal-like only once and using it consistently. Similarly, ensure consistent abbreviations for terms like LAR, IM, MES, etc.
4. Remove the content specified in Lines 294-295.
5. Fill in the missing information in Lines 303 and 312; specify the required details.

Reviewer 2 Report

Comments and Suggestions for Authors

Major changes: 

1. Line 81- you state that demographics are in Table 4, but Table 4 is you antibody table and there does not appear to be a table in the manuscript that summarizes the demographics.  

2. Table 2- This table is not clear.  It is unclear and I assumed that (Mean (SD) and (N (%) indicated the numbers in the last two columns? Why are some of the % numbers in parentheses indicate that they are negative? I think it would be helpful to have column headings and to have all of the Mean (SD) all at the top so move the Ki67 to the top, then put in a new set of column headings to indicate N(%) for the rest of the table.  Also why are you indicating the abbreviations for Basal-like (BL), immunosuppressed (IS), and immunoactivated (IA) in the middle of the table when these abbreviations are not even used anywhere else in the table? In the legend it needs to be indicated for AC, TXT, and wPac that the x4 and x12 means "number of cycles" like you mention in Table 3.  

3.  Line 88- What 4 markers did you stain for and why? This needs to be discussed.  

4. I originally thought you were not going to discuss Figure 1C & D until I read it after reading the data for Figure 2 in lines 141-145.  I would suggest moving Lines 141-145 to line 123 so it stays with the rest of Figure 1.  

5. Figure 1 needs a legend

6. Figure 1- indicate whether your time is in months or weeks on the x-axis.

7. Line 130- Is this the incorrect table that goes with this graph?   

8. Line 133- You state that the pattern of recurrence is graphically distinct but you should also indicate that the data is also not significant though based on the p value for the Log rank test.  

9. Figure 2- I would recommend putting the LAR into C & D even though it is zero, just so the data is still there and you don't have to depend on reading about this in the text.  

10. Table 5- there is no legend, M or I abbreviations are not explained, Weak (W) could be moved to the legend so W can just be used in the Table.  

11. Supplementary figure 1 is not discussed in the Results section.  

Minor changes:

1. pCR abbreviation is stated in line 15, you don't need to restate this in line 21, just use the abbreviation.  

2. Line 47-51: The LAR abbreviation is repeated and the mesenchymal abbreviation is different in line 47 vs. line 51, probably better to just go with MES since you use this in the rest of the manuscript instead of just M.  

3. Table 1. Legend- You use MES for mesenchymal stem-cell like, you should change this to MSL to stay consistent to line 48 and so it is not confused with just the MES used for mesenchymal alone from line 51.  Also you have just M for mesenchymal but use MES throughout the manuscript this needs to be changed to be consistent.  

4. Table 3 legend- W- weak, I- intense are given as abbreviations but W and I do not appear anywhere in the table, are these missing? 

5. Line 108- what does CI stand for? 

6. Line 108, 111- the P in PCR needs to be lower case pCR.  

7. Line 116- remove e in meDFS, so it is mDFS

8. Line 117- ICH needs to be changed to IHC.  

9. Figure 1B & D, Line 126, 131, needs lower case P for pCR.  

10. Line 152- add a comma after Oakman and periods after C and al.  

11. Line 153- remove the extra "l" for et al.  

12. Line 163- remove the parentheses around 15.0% as it is apart of the sentence and not a side note.  

13. Line 187- receptor is already apart of the abbreviation for AR so you need to remove "receptor" after AR.  

Round 2

Reviewer 1 Report

Comments and Suggestions for Authors

The authors have satisfactorily addressed the comments raised in the initial review, except regarding the technical hurdles linked to standardizing IHC protocols for diagnostic protocols, including their validation and approval by regulatory bodies. Please mention, and consider briefly highlighting the advantages of leveraging the digital tools currently accessible to aid in pathological assessments.

Comments on the Quality of English Language

Typos have been introduced: and a the patients; receptor.Staining; assocaited

Regarding presentation:

1. use the correct referencing style of the Journal. adapt references 1112131415161718 as [11-18]

2.  In their rebuttal letter, the authors state "We have added in the line 246 that the staining was multiplex", whereas line 273 indicates now "singleplex IHC staining results". Be consistent 

Author Response

For research article

Triple negative breast cancer subclassified by immunohistochemistry: correlation with clinical and pathological outcomes in patients receiving neoadjuvant chemotherapy.

Response to Reviewer 1 Comments (round 2)

1. Summary

2. Questions for General Evaluation

Reviewer’s Evaluation

Response and Revisions

Does the introduction provide sufficient background and include all relevant references?

Yes

[Please give your response if necessary. Or you can also give your corresponding response in the point-by-point response letter. The same as below]

Are all the cited references relevant to the research?

Yes

Is the research design appropriate?

Can be improved

Are the methods adequately described?

Yes

Are the results clearly presented?

Can be improved

Are the conclusions supported by the results?

Yes

3. Point-by-point response to Comments and Suggestions for Authors

Comments 1: The authors have satisfactorily addressed the comments raised in the initial review, except regarding the technical hurdles linked to standardizing IHC protocols for diagnostic protocols, including their validation and approval by regulatory bodies. Please mention, and consider briefly highlighting the advantages of leveraging the digital tools currently accessible to aid in pathological assessments.

Response 1: Thank you for pointing this out. We agree with this comment. Therefore, we have introduced alterations on the latest paragraph from the discussion as follows:

Although the results of this study are promising, they should be interpreted in light of these limitations. Our retrospective design imposes challenges in survival analysis including selection and recall bias. The eventual changes in TNBC treatment protocols over the study period could introduce a confunding variable to the outcomes evaluated as well as other variables not analyzed such as comorbidities and genomic background. In case implemented, our proposed IHC panel will eventually increase in cost to current standard of practice recommendation and could eventually cause delay in standard of practice results delivery. Moreover, there are several technical hurdles linked to standardizing IHC protocols for diagnostic protocols, including their validation and approval by regulatory bodies, which would be required before the panel we proposed is considered outside of a research environment. There is no consensus about the use of an IHC subtyping panel, and IHC staining was performed utilizing archival tumor tissue from 2008 onward, which could have impacted the performance of the assay and requires validation using an established method for intrinsic genomic subtyping

Comments 2: Typos have been introduced: and a the patients; receptor.Staining; assocaited

Response 2. We agree and amended the highlighted typos on the new version attached.

Comments 3: Regarding presentation:

1.     use the correct referencing style of the Journal. adapt references 1112131415161718 as [11-18]

2.     In their rebuttal letter, the authors state "We have added in the line 246 that the staining was multiplex", whereas line 273 indicates now "singleplex IHC staining results". Be consistent 

Response 3: Agree. We have:

a.     Amended the referencing as recommended

b.     Would like to clarify our typo on the previous rebuttal letter and to reassure the reviewed that the staining was singleplex

We hope the revised version covers the reviewers’ suggestions and look forward to hear the feedback in due course.